# Effect of Regular Electrotherapy on Spinal Flexibility and Pain Sensitivity in Patients with Chronic Non-Specific Neck Pain and Low Back Pain: A Randomized Controlled Double-Blinded Pilot Trial

**DOI:** 10.3390/medicina59050823

**Published:** 2023-04-23

**Authors:** Asami Naka, Clea Kotz, Edith Gutmann, Sibylle Pramhas, Regina Patricia Juliane Schukro, Robin Ristl, Othmar Schuhfried, Richard Crevenna, Sabine Sator

**Affiliations:** 1Department of Special Anaesthesia and Pain Medicine, Vienna General Hospital, Medical University of Vienna, 1090 Vienna, Austria; 2Section for Medical Statistics, Center for Medical Data Science, Medical University of Vienna, 1090 Vienna, Austria; 3Department of Physical Medicine, Rehabilitation and Occupational Medicine, Vienna General Hospital, Medical University of Vienna, 1090 Vienna, Austria

**Keywords:** chronic pain, neck pain, low back pain, electrotherapy, range of motion

## Abstract

*Background and Objectives*: Chronic neck pain and low back pain are common conditions in high-income countries leading to social and medical problems such as invalidity and decreased quality of life. The aim of this study was to investigate the effect of supra-threshold electrotherapy on pain level, subjective feeling of disability, and spinal mobility in patients with chronic pain in the spinal cord. *Materials and Methods*: 11 men and 24 women with a mean age of 49 years were randomly divided into three groups: group 1, “therapy”: supra-threshold electrotherapy was applied on the whole back after electrical calibration; group 2, “control”: electrical calibration without successive electrotherapy; group 3, “control of control”: no stimulation. Sessions were performed once a week and six times in total, each lasting 30 min. The numeric pain rating scale (NRS), cervical and lumbar range of motion (ROM), as well as disability in daily live were investigated before and after the sessions using questionnaires (Neck Disability Index, Roland Morris Questionnaire, Short-form Mc Gill Pain Questionnaire (SF-MPQ)). *Results*: Spinal mobility improved significantly in the lumbar anteflexion (baseline mean, 20.34 ± SD 1.46; post session mean, 21.43 ± SD 1.95; *p* = 0.003) and retroflexion (baseline mean, 13.68 ± SD 1.46; post session mean, 12.05 ± SD 1.37; *p* = 0.006) in the group receiving electrotherapy. Pain levels measured by the NRS and disability-questionnaire scores did not differ significantly before and after treatment in any of the groups. *Conclusions*: Our data indicate that regular supra-threshold electrotherapy for six times has a positive effect on lumbar flexibility in chronic neck pain and low back pain patients, whereas pain sensation or subjective feeling of disability remained unchanged.

## 1. Introduction

Chronic neck pain and low back pain are common causes of invalidity and decrease in quality of life in high-income countries. However, the effectiveness of pain medication is limited [1]. Therefore, alternative therapeutic approaches are gaining importance.

Electrotherapy, especially transcutaneous electrotherapy (TENS), has been an inexpensive, easily applied and widely used tool in different painful conditions for several decades. The reduction in pain by electrotherapy is suggested to be due to several mechanisms. One of them is the gate-control-theory by Melzack and Wall, which proposes the modulation of afferent nociceptive patterns in the superficial dorsal horn of the spinal cord, i.e., the inhibition of small afferent nociceptive fibres by large afferent fibres, and consequently, activation of inhibitory interneurons, thus leading to less nociceptive input to neurons projecting to the brain [2]. Other suggested explanations for the decrease in pain sensation after TENS are the activation of descending inhibitory pathways in the central nervous system by effects mediated by endogenous opioids [3], or a reduction in the blood level of proinflammatory cytokines such as IL-1, lL-6 and TNFα [4].

However, up to now there is scarce evidence for the clinical efficacy of TENS therapy in neck pain [5] or low back pain [6], or for the effectiveness of other electrotherapies in neck pain [7]. Controversial results concerning functional disability or quality of life are reported by previous randomized controlled trials analyzing the clinical effectiveness of TENS. 

Therefore, the aim of the present study is to investigate the effect of regular, supra-threshold electrotherapy in pain sensation and in cervical and lumbar range of motion, as well as in the subjective feeling of disability in daily life. Thereby, a mat covering the whole spinal cord was used for current application, aiming a deeper tissue penetration and an increased effectiveness compared with conventionally used TENS.

## 2. Materials and Methods

### 2.1. Study Design

This randomized controlled, double-blind pilot study was performed at the Medical University of Vienna, Department of Special Anesthesia and Pain Therapy. 

### 2.2. Ethics and Clinical Registration

The study was approved by the local ethics committee of the Medical University of Vienna (registration number: EK Nr. 1571/2014, serial number 456499) and registered with the Austrian Federal Office for Safety in Health Care BASG (INS-621000-0687), as well as with ClinicalTrials.gov, accessed on 5 January 2023 (INS-621000-0687-002).

### 2.3. Study Population

A total of 54 male and female patients aged 18 years or over with chronic neck pain or low back pain were enrolled in the study. The inclusion criteria were a minimum numeric rating scale equal to or greater than five, and pain duration of more than three months. The oral pain medication had to be stable at least four weeks prior to the first investigation. No additional pain therapies (e.g., acupuncture, physical therapy, osteopathy, intravenous pain therapy, intramuscular or subcutaneous injection of cortisone or local anesthetics) were allowed parallel to the study. Such pain therapies had to be discontinued at least four weeks before the study onset. Patients with experience in TENS were excluded. Other exclusion criteria were pregnancy, epilepsy, cardiac arrhythmia, cardiac surgery including implanted pacemaker or defibrillator, previous surgery in the spinal cord, infectious diseases of the spinal cord, malignant tumors with or without secondary blastomas in the spinal cord, severe radicular pain with acute paralysis in the extremities, or an ongoing pension application. 

### 2.4. Study Process

Written informed consent was obtained from all patients included in the study. Before the initiation of the electrotherapy sessions and recording of demographic information including age, gender, weight, height and body mass index, laboratory parameters and an X-ray of the spinal cord were examined to exclude any signs of new onset malignancies or infections in the spinal cord. Before and after the therapy sessions, actual pain in rest and under activity, as well as the maximum and minimum pain in the last four weeks were assessed using the numeric rating scale (NRS) for the cervical, dorsal, and lumbar region, respectively. The quality of pain was evaluated using the Short-form Mc Gill Pain Questionnaire (SF-MPQ). The PainDETECT test was used to screen for radicular involvement. Range of motion (ROM) was assessed using the Cervical Range-of-Motion Instrument (CROM^®^; cervical range of motion CROM basic spinal assessment device; ProHealthcareProducts.com, accessed on 5 January 2023, 770 East Main Street, #201 Lehi, UT 84043, USA) and with the modified Schober method (assessment of anteflexion and retroflexion 10 cm above and 5 cm below the anterior superior iliac spine). Subjective feeling of disability in daily life was assessed with questionnaires (Neck Disability Index for cervical pain, and Rolland Morris Questionnaire for low back pain). 

Therapy sessions were conducted weekly with a duration of 30 min each and six times in total. All the patients lay supine on an electrotherapy mat covering the whole spinal cord (StimaWELL^®^ 120MTRS; Schwa-medico, Wetzlarer Straße 41–43, 35630 Ehringshausen, Germany) that was warmed to 25 °C (see Figure 1).

### 2.5. Randomization and Blinding

Computer-based randomization stratified by age, gender and pain location (neck pain vs. dorsolumbar pain) was performed. All patients, as well as the physician performing the physical examination, were blinded and randomly divided into three groups. The first group of patients (“therapy”) received a supra-threshold electrotherapy with alternating current application, consisting of high-frequency stimulation at 100 Hz and low-frequency stimulation at 2 Hz, which is used in daily clinics [8] and been shown to elicit a synergistic release of different endogenous opioid peptides [9]. This stimulation was applied for 30 min after the electrical calibration of the StimaWELL^®^ mat at the very beginning of the sessions. The aim of the calibration was to determine the individual stimulation threshold and consisted of short electrical pulses applied in each segment and in each side of the back for few seconds, respectively. Starting with subthreshold stimulation, the intensity was slowly increased until stimuli were realized but still felt as non-painful. In the second group of patients (“control”), the device calibration was performed without any further subsequent electrical stimulation. The third group (“control of control”) lay on the mat without the application of any electrical stimulation.

### 2.6. Outcome Measurements

The primary endpoint of the study was the pain sensation (NRS) of the chronic neck pain and low back pain after electrotherapy compared with the control groups. The secondary endpoints were QoL, assessed by the Neck Disability Index and the Rolland Morris Questionnaire, as well as the ROM measured with CROM^®^ and the modified-Schober method.

### 2.7. Statistical Analysis

Statistical analysis was performed using Sigma Plot 12.0 (Systat Software GmbH, Schimmelbuschstraße 25, 40699 Erkrath, Germany). The intra-individual analysis (i.e., comparison between baseline values and values after 6 electrotherapy sessions), as well as inter-individual analysis (i.e., comparison between the three groups) were performed using the paired *t*-test or the One-way ANOVA for normal distribution, and Wilcoxon Signed Rank test or Kruskal–Wallis ANOVA on ranks, if the normality test (Shapiro–Wilk test) failed. All comparisons in this pilot study were regarded to be of exploratory nature and no adjustment for multiple testing was performed. *p* < 0.05 was considered as statistically significant. 

## 3. Results

A total of 54 patients with chronic neck pain or low back pain were included in the study. Thereof, 35 patients (11 men, 24 women) were finally analyzed. Among these patients, seven patients had cervical pain only, 14 patients had low back pain solely, and 14 patients had both. All patients primarily exhibited local back pain, without or with little radicular pain. 

The subset of 21 patients with chronic neck pain (i.e., seven patients with cervical neck pain and 14 with both neck- and low back pain) were divided into the electrotherapy group (*n* = 9), control group (*n* = 7), and control of control (*n* = 5). The subset of 28 patients with chronic low back pain were further divided into the therapy group (*n* = 10), control group (*n* = 10), and control of control (*n* = 8) (see Figure 1). 

For the demographic description of the patients, see Table 1 and Table 2. There was no significant difference in age and BMI between the groups.

Among the 35 patients, 12 (34.29%) did not take any pain medication. Of the remaining 23 patients, 22 took non-opioid medication regularly, 6 of whom took a combination with opioids and 8 of whom took antidepressant medication. One patient took antidepressants only. Depression was previously diagnosed in 7 patients out of 35 (1 patient in control in control; 4 in the control group; 2 in the therapy group; 20% in total). No increase or decrease in the intake of rescue pain medication was reported by any of the patients investigated during or after the weekly sessions.

The average, minimum and maximum NRS of cervical and low back pain, at rest and under activity showed no significant changes before and after the therapy sessions in any of the groups. (The mean and standard deviation of baseline (pre) and post session (post) values are stated below.)

### 3.1. Neck Pain

Average NRS: therapy: pre 6.2 ± 2.3, post 5.3 ± 2.45, *p* = 0.46; control: pre 4.7 ± 1.5, post 4 ± 1.9, *p* = 0.41; control of control: pre 5.6 ± 2.41, post 5.4 ± 1.95, *p* = 0.8;

Minimum NRS: therapy: pre 3.33 ± 3.35, post 2.89 ± 1.9, *p* = 0.72; control: pre 1.71 ± 1.7, post 2 ± 1.53, *p* = 0.63; control of control: pre 2.6 ± 2.97, post 1.4 ± 2.19, *p* = 0.32;

Maximum NRS: therapy: pre 8.22 ± 1.2, post 7.56 ± 1.67, *p* = 0.24; control: pre 6.71 ± 0.76, post 6.14 ± 1.68, *p* = 0.44; control of control: pre 7.2 ± SD 1.48, post 7.6 ± 1.82, *p* = 0.76;

NRS at rest: therapy: pre 5.11 ± 2.85, post 4.44 ± 2.52, *p* = 0.58; control: pre 4.14 ± SD 2.41, post 2.43 ± 1.27, *p* = 0.04; control of control: pre 4.2 ± 3.9, post 4.2 ± 3.35, *p* = 1;

NRS under activity: therapy: pre 5.4 ± 2.55, post 4.67 ± 2.65, *p* = 0.46; control: pre 5.43 ± 1.99, post 3 ± 2, *p* = 0.04; control of control: pre 4.6 ± 2.07, post 5.4 ± 1.95, *p* = 0.18.

### 3.2. Low Back Pain

Average NRS: therapy: pre 5.5 ± 1.58, post 6.2 ± 2.15, *p* = 0.17; control: pre 4.5 ± 0.97, post 4.7 ± 2.91, *p* = 0.1; control of control: pre 4.88 ± 2.36, post 4 ± 2.56, *p* = 0.06;

Minimum NRS: therapy: pre 1.9 ± 2.02, post 3 ± 2.31, *p* = 0.19; control: pre 1.5 ± 1.65, post 2.3 ± 2.31, *p* = 0.29; control of control: pre 2.38 ± 2, post 2.13 ± 2.23, *p* = 1;

Maximum NRS: therapy: pre 8.5 ± 1.27, post 8.1 ± 1.52, *p* = 0.25; control: pre 6.6 ± 1.51, post 4.7 ± 2.91, *p* = 0.12; control of control: pre 7.75 ± 1.83, post 6.63 ± 2.92, *p* = 0.11;

NRS at rest: therapy: pre 3.8 ± 2.44, post 3.9 ± 3.14, *p* = 0.85; control: pre 3.7 ± 2.31, post 3 ± 3.02, *p* = 0.47; control of control: pre 3.25 ± 2.66, post 3.38 ± 4.14, *p* = 0.91;

RS under activity: therapy: pre 5.8 ± 2.1, post 5.3 ± 2.5, *p* = 0.59; control: pre 4.1 ± 2.73, post 3.44 ± 2.74, *p* = 0.59; control of control: pre 5.5 ± 3.63, post 4 ± 3.92, *p* = 0.33 (see Figure 2).

Next, a comparison of the total NRS at rest, under physical activity, minimal, maximal, as well as average NRS between the three groups was performed. Thereby, the respective sums of the NRS of the cervical, dorsal, and lumbar regions were calculated. ANCOVA using the pre-value of the groups as a covariant did not reveal any significant differences between the means post intervention (*p* > 0.05, respectively). The mean and standard deviation of the baseline (pre) and post session (post) of all the groups, as well as the 95% confidence intervals (CI) are stated, as follows:

Average NRS: Pre: therapy (*n* = 13): 13.15 ± 6.04; control (*n* = 13): 7.31 ± 3.82;

control of control (*n = 9*): 9.22 ± 4.82. Post: therapy (*n* = 13): 13.2 ± 5.7; 

control (*n* = 13): 6.8 ± 4.8; control of control (n = 9): 9.1 ± 6.6. 

Comparison therapy vs. control: CI of the mean difference 2.02 (−1.54–5.58),

*p* = 0.25.

Comparison therapy vs. control of control: CI of the mean difference 0.52 (−2.71–3.74),

*p* = 0.74.

Minimum NRS: Pre therapy (n = 13): 5.0 ± 5.0; control (*n* = 13): 2.62 ± 3.48; control of control (*n* = 9): 4.44 ± 4.61. Post: therapy (*n* = 13): 5.5 ± 4.4; control (*n* = 13): 3.5 ± 3.3; control of control (*n* = 9): 2.8 ± 3.6.

Comparison therapy vs. control: CI of the mean difference 0.52 (−1.94–2.97), *p* = 0.67.

Comparison therapy vs. control of control: CI of the mean difference 2.43 (−0.33–5.19), 

*p* = 0.08.

Maximum NRS: Pre: therapy (*n* = 13): 18.62 ± 6.29; control (*n* = 13): 10.69 ± 4.13; control of control (*n*= 9): 14.56 ± 6.64. Post: therapy (*n* = 13): 18.8 ± 5.4; control (*n* = 13): 9.8 ± 5.4; 

control of control (*n* = 9): 13.7 ± 6.4.

Comparison therapy vs. control: CI of the mean difference 4.57 (−0.1–9.25), *p* = 0.055.

Comparison therapy vs. control of control: CI of the mean difference 2.9 (−1.57–7.38),

*p* = 0.19.

NRS at rest: Pre: therapy (n = 13): 8.77 ± 5.97; control (*n* = 13): 6.15 ± 5.03; control of control (*n* = 9): 6.67 ± 5.72. Post: therapy (n = 13): 7.7 ± 5.5; control (*n* = 13): 4.8 ± 3.6; 

control of control (n = 9): 7.0 ± 7.2.

Comparison therapy vs. control: CI of the mean difference 1.38 (−1.46–4.23), *p* = 0.32.

Comparison therapy vs. control of control: CI of the mean difference −1.01 (−4.81–2.79),

*p* = 0.58.

NRS under physical activity: Pre: therapy (n = 13): 12.54 ± 5.77; control (*n* = 13): 

7.31 ± 5.54; control of control (*n* = 9): 9.44 ± 5.88; Post: therapy (*n* = 13): 11.6 ± 6.0; 

control (*n* = 12): 5.6 ± 4.5; control of control (*n* = 8): 9.6 ± 7.4.

Comparison therapy vs. control: CI of the mean difference 3.12 (−1.27–7.5), *p* = 0.15.

Comparison therapy vs. control of control: CI of the mean difference −0.96 (−6.44–4.51),

*p* = 0.72.

No improvement in daily living was shown either in the Neck Disability Index or in the Rolland Morris Questionnaire. Likewise, there were no significant differences in the total scores of the SF-MPQ (see Table 3).

Regarding the cervical range of motion, in all groups, there were no significant changes in any planes assessed before and after the electrical stimulation (see Table 4).

However, regarding the ROM in the lumbar region, there was a significant improvement in comparison with the baseline values in the group that received electrotherapy, in both anteflexion (therapy: pre: mean 20.34 ± SD 1.46, post: mean: 21.43 ± SD 1.95, *p* = 0.003; control: pre 21 ± 1.65, post 20.63 ± 1.35, *p* = 0.25; control of control: pre 21.84 ± 1.83, post 21.43 ± 2.42, *p* = 0.39) as well as in retroflexion (therapy: pre 13.68 ± 1.46, post 12.05 ± 1.37, *p* = 0.006; control: pre 13.58 ± 1.08, post 13.74 ± 0.73, *p* = 0.82; control of control: pre 12.67 ± 1.14, post 12.71 ± 1.11, *p* = 0.9; see Figure 3).

To assess the effectiveness of blinding, patients were asked at the end of the study whether they had received true or sham therapy. Among 25 subjects, 12 (48%) were correct (therapy: *n* = 1; sham: *n* = 11). Among the 13 patients that were incorrect, 8 patients thought they had received sham therapy, although regular electrotherapy was applied. The remaining five patients that wrongly believed they had received verum therapy were all assigned to the group “control of control”, where no electrical stimulation was applied at all.

## 4. Discussion

The current pilot study shows a statistically significant improvement in the mobility of the lower back in patients with chronic back pain using an easy-handling and time-saving electrotherapeutical device. However, there was no statistically significant amelioration in the pain sensation. Of note, due to the exploratory nature of the study, these findings should be considered to be hypothesis generating. At first sight, this increase in ROM of the lower back appears unspectacular with poor means because subjective pain levels remained unchanged. However, clinical experience has shown that physical integrity is as important as an amelioration of subjective feeling of pain. Poor physical activity can aggravate chronic pain, e.g., by increasing myogelosis or joint stiffening that results from maintaining a protective posture constantly, and functional mobility is in close connection with subjective well-being and quality of life in chronic pain patients, most likely due to its impact on the personal range of activity and resulting positive influence on the patient’s mental state and self-esteem. Moreover, previous literature has described that chronic pain, depression and cardiovascular diseases can co-occur [10,11]. Thus, an increase in spinal range of motion by itself might prevent progressive immobility and avoid comorbidities associated with poor mobility such as obesity, aggravation of diabetes or cardiovascular diseases. 

The significant increase in the lumbar ROM we observed after regular electrotherapy contrasts with the results of Deyo and colleagues who reported no functional amelioration using TENS [12], but it is in concordance with the findings of Rajfur et al. who reported an increase in functional abilities using TENS, interferential current stimulation and diadynamic current [13]. One explanation for the discrepancy of the results could be, as already stated by Rajfur and colleagues, a possible deeper tissue penetration of the electrical stimulation by using a bigger area instead of small patches, possibly leading to better tissue vascularization and greater reductions in myofascial contraction and blockade by using mid-frequent electrotherapy. Interestingly, Rajfur and colleagues not only observed functional amelioration but also a significant pain reduction in all the groups receiving electrotherapy, including low-frequent electrotherapy (i.e., TENS, high-voltage electrical stimulation, diadynamic currents), which contrasts with our data. Nevertheless, patients receiving interferential therapy had the greatest pain relief. The highest functional improvement was also observed in the group receiving mid-frequent interferential therapy, possibly due to the deeper current penetration into the tissue, as stated by the authors.

As mentioned above, we did not observe any significant difference in the pain sensation nor in the quality of life after electrotherapy in any of the tests performed. Thereby, several technical considerations concerning the weekly therapy sessions are to be noted: The therapy protocol including the duration of electrical stimulation, session interval and total number of therapies could be still inadequate for long-lasting pain relief. Some patients who received electrotherapy stated that, although the stimulation felt adequate in the beginning, it became too strong after some time, and the pain condition became worse afterwards. Other patients reported a short amelioration of the pain state right after the electrotherapy sessions. However, such pain reduction mostly lasted only a few hours and disappeared within the same day. An intra-individual adaptation of the stimulation, or an inter-individual adaptation of the session interval might be possibly beneficial.

In addition, a greater improvement in mobility or pain relief could be possibly achieved if electrotherapy is combined with heat application since it is known that a multimodal therapy approach can be more effective concerning pain relief [14,15]. The StimaWELL^®^ device can indeed apply heat in addition to electrotherapy. Nevertheless, to keep the study protocol simple, the mat was warmed at room temperature only and no additional heat was applied. 

Furthermore, an optimal skin contact is crucial. In a few cases, it was difficult to find the adequate mat positioning. If the contact of the mat and the spinal cord was insufficient and if a higher simulation threshold was required, cushions were put under the mat or under the knee to optimize the skin contact. However, finding the appropriate position in the cervical region was still difficult with some of the patients.

Nevertheless, the electrotherapeutic mat that was utilized in this study is useful in daily clinical practice due to its easy handling. It could even be operated by the patients themselves after a short training, and thus save much time in daily clinics, where time is scarce anyway.

The placebo effect is a well-known problem in studies investigating patients with chronic pain [16]. Positive or negative suggestions are reported to influence the subjective sensation of chronic pain. A previous study reported reductions in low back pain in the presence of solicitous partners, whereas pain sensation was not significantly changed in the presence of non-solicitous spouses [17]. Another study investigated the effect of verbal suggestion on post needling soreness [18]. In this acute pain model, the authors observed no significant pain amelioration after positive suggestion; one explanation for the negative results could be that verbal suggestion may have a more obvious effect in chronic pain states, which is often paired with higher anxiety levels compared with acute pain. Indeed, Fernández-Carnero et al. observed a higher probability of success of a cervical mobilization technique in patients with chronic neck pain and reduced lateral flexion who showed higher anxiety levels [19]. Another possible explanation for the reduced effectiveness of verbal suggestion in acute pain compared with chronic pain might be that chronic pain patients may have a greater desire for pain amelioration and for situational changes; previous studies have reported that patients’ positive health care experiences in the past, positive training expectations, as well as the possibility of self-management (e.g., learning the disease pathology and treatment options) may have beneficial effects on chronic low back pain [20]. Other potentially pain-influencing factors are, e.g., patients’ beliefs in illness or therapy, their professional education and experience, and the empathy or beliefs of health care providers [21], as well as the relationship between patients and practitioners [22,23]. Hence, in our study, everybody in the team was instructed to minimize contact with the patients during the weekly sessions to avoid verbal suggestion. All the patients were led into the investigation room by nurses in our pain out-patient clinic, asked to take off their clothes and to lay on the electrotherapy mat. Then, the nurses left the room. After the sessions they escorted the patients out. The physician adjusting the electrical stimulation asked the patients whether the stimuli were appropriate, covered the screen of the device with a towel so that activation or inactivation of the device could not be seen by the patients, and then walked out of the room. Doors between the investigation rooms were not completely shut so that patients could call for help if there was any discomfort. However, the physical effects of small conversations cannot be completely excluded.

The degree of blinding was assessed by asking the patient which treatment they thought was administered (treatment vs. control vs. control of control). Indefinite answers were not accepted so that patients had to guess if they were uncertain. The proportion of patients who answered correctly was 48%. Considering that the probability of guessing correctly by chance is about 33%, and that in the case of unblinding or failed blinding, the percentage of correct answers would be 100%, our result seems, at first sight, not that unsuccessful. However, this consideration is statistically incorrect, and a calculation such as the blinding index proposed by James and colleagues [24] or Bang and colleagues [25] could not performed since the proportion of patients giving a guess with the answer “I don’t know” was not recorded and thereby missing in the formula. Thus, it cannot be stated whether our blinding method was successful or not, and an influence on the dropout rate, the subjective pain states, or the activity in daily living cannot be excluded. Nevertheless, we suggest that the significant improvement in the ROM of the lower back that we observed after electrotherapy is little influenced by unblinding.

It is obvious that the number of the participants is small, and a greater number of patients would improve the statistical power. Initially, sample size was calculated based on a previous TENS study performed in patients with chronic low back pain, where a difference of NRS of 1 between the groups (primary end point) was used. A power of 80% and standard deviation of 1.5 were set based on the study of Pop et al. [26]. A sample size of *n = 37* was calculated. Taking a dropout rate of 30% in consideration, the optimal sample size was increased to *n* = 50 for the intervention group and control group, respectively. The patient number of the control of control group was set to 10 because no physiological changes were expected in these patients who did not receive any electrical current. However, because of difficulties in patient recruitment (there were hardly any chronic pain patients who had not experienced TENS previously) and due to lack of time and resources, there were finally fewer patients included in the study. This led to more varied distribution of patients than initially planned; however, there were no statistically significant differences in any of the baseline values between the groups.

Furthermore, insufficient blinding or biasing due to dropouts might falsely produce a negative effect of electrotherapy. Most patients (14/19) who dropped out did so right after the baseline investigation before randomization was performed and the weekly sessions started. The remaining five patients belonged to the group receiving supra-threshold electrotherapy and quit after the performance of one to four therapy sessions (two patients after the first session, one patient after the second session, one patient after the third session, one patient after the fourth session) No adverse events were observed in these patients. Since the patients were not obliged to state a reason for the discontinuation, this could not be further analyzed. In two of the five patients, depression was diagnosed, and one patient had a diagnosis of depression and anxiety disorder before the enrolment. NRS, QoL and ROM were investigated at the very beginning of the study before randomization was performed, and at the end of the study after the completion of the six sessions of intervention; therefore, we cannot determine whether there were any changes in the pain level between these times. A possible reason for discontinuation may have been a perceived inefficacy of the therapy and a mismatch with the patients’ therapeutic beliefs and expectations.

Finally, another possible explanation for the unchanged pain sensation could be that, according to the bio–psycho–social model of pain, the existence of psychologic or socioeconomic factors that do not change easily, such as anxiety, maladaptive behavior, stressful circumstances, poverty, or biological factors, such as remodeling in the central nervous system, tend to result in chronic pain states [27]. Since central nervous system remodeling has been observed after the performance of psychotherapy, biofeedback or meditation [28], it could be interesting to monitor anxiety levels before and after the intervention, and to assess whether the combination of electrotherapy and behavioral therapy leads to both clinical benefit and a change in central nervous nociceptive circuits in the long term. In the current study, validated tests were not used to check for mental health problems, and socioeconomic factors were not examined in detail either. These remain questions for future studies.

## 5. Future Directions and Clinical Implications

In this study, the use of mid-frequency electrotherapy was restricted to chronic neck pain and low back pain. However, Stimawell^®^ can also be used as a part of physiotherapy to strengthen the back muscles, e.g., in frailty due to chronic critical illness or in postoperative settings. Its easy handling with an attached remote controller allows patients to manage their pain therapy concerning the optimal timing, frequency, and stimulus intensity themselves. In combination with optimal dietary nutrition, it might be an effective and time-efficient tool in rehabilitation programs, which is a topic for future studies. The reported values for mean and standard deviations may be used in the sample size planning of future studies.

## 6. Conclusions

The current pilot study shows that regular mid-frequency electrotherapy applied in patients with chronic neck pain or non-specific low back pain improved the range of motion of the lower spinal cord significantly, whereas pain sensation remained unchanged. Although a significant improvement of pain levels could not be revealed in this trial, the electrotherapy mat remains a feasible therapeutic option as functional improvement also constitutes an important therapeutic goal in chronic low back pain.

## Data Availability

Data and results are archived on the online database “Clincase” (https://akimstud.meduniwien.ac.at/clincase/app (accessed on 5 January 2023)), which is managed by the IT systems and communication of the Medical University of Vienna.

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
