# Peer review of "Effect of Regular Electrotherapy on Spinal Flexibility and Pain Sensitivity in Patients with Chronic Non-Specific Neck Pain and Low Back Pain: A Randomized Controlled Double-Blinded Pilot Trial"

_medicina, 2023, doi:10.3390/medicina59050823_

Round 1
Reviewer 1 Report
I'm sorry, but I have a lot of questions about the process and data analysis of this study. Many, many things seem problematic to me. The authors describe a computerised randomisation But
- Why then are the groups far different in size?
-Why are the pain locations unequally distributed?
-I am particularly lacking an explanation here for the strongly unequal group distributions.
I could not find a determination of the optimal sample size Why not?
Is there an ethics vote Why not?
A variety of statistical test procedures are identified No information on measures against alpha error inflation What is being done about the problem of multiple testing?
No effect sizes are reported for any statistical group comparison. Why not Especially since this is a pilot study, the effect sizes should be given
For example, I could not find degrees of freedom in group comparisons. Why were neither the test values nor the degrees of freedom given?
Why did "only" five patients in the intervention group refuse to participate? This would have to be discussed intensively.
Since the groups seem to differ significantly on the pre-values, the pre-values should be considered as covariates in the context of an analysis of variance. In the pairwise comparisons presented in the manuscript, the pre-value is not taken into account in the analysis of the post-values.
Author Response
Response to Reviewer 1 Comments
Point 1: Why then are the groups far different in size? Why are the pain locations unequally distributed? I am particularly lacking an explanation here for the strongly unequal group distributions.
Response 1: Thank you for pointing to this issue. This study was initially not planned to be a pilot study, but a study with an intervention group with 50 patients with chronic back pain and/or neck pain, 50 patients in the control group with device calibration but without electrotherapy, and 10 patients neither with device calibration nor electrotherapy.
The sample size was calculated based on previous TENS-studies performed in patients with chronic low back pain, where a difference of NRS of 1 between the groups (primary end point) was used for sample size calculation. Power of 80 % and standard deviation of 1.5 were set based on the study of Pop et al. Taking a dropout rate of 30 % in consideration, the calculated sample size of n = 37 was increased to n= 50 for the intervention group and control group, respectively. N-number of the control of control was set to 10, because no physiological changes were expected since no current was applied to these patients.
The randomizer (statistical program) was set to group the patients equally depending on sex, pain location (cervical pain vs. dorsolumbar pain), as well as to different age classes (18-40 years, 41-60 years, 61-80 years, over 80 years).
Because of difficulties in patient recruitment (there were hardly any chronic pain patients who did not experience any kind of electrotherapy previously) and due to lack of time (we stated in the ethical application to finish the study after 2 years), there were finally less patients included and the patient distribution more various than initially planned.
It is now added in the revised version of the manuscript in line 449-462.
Point 2: I could not find a determination of the optimal sample size Why not?
Response 2: Please see Response 1 above and line 449-462 in the revised manuscript.
Point 3: Is there an ethics vote Why not?
Response 3: There is a positive vote from the ethic committee from the Medical University of Vienna (EK Nr.: 1571/2014). Approval is stated in the text in line 523-527. Please also follow the link of the home page of the MUW ethic committee below:
https://ekmeduniwien.at/core/catalog/2014/
We have added the ethics part also in the methodology as “2.2. Ethics and Clinical Registration”, (line 76-80).
Point 4: A variety of statistical test procedures are identified No information on measures against alpha error inflation What is being done about the problem of multiple testing?
Response 4:
Thank you for pointing to the problem of multiple testing. Of note, the study was planned for one primary comparison of average NRS between electrotherapy and sham treatment. Further comparisons were regarded as exploratory or hypothesis generating. However, given the role as pilot study the study should in general be considered to be of exploratory nature. We clarify this point in the Methods Section, saying: “All comparisons in this pilot study were regarded to be of exploratory nature and no adjustment for multiple testing was performed.” In the Discussion of the revised manuscript we point out more clearly that the results are to be interpreted in an exploratory manner, by adding a sentence saying “Of note, due to the exploratory nature of the study, these findings should be considered to be hypothesis generating”.
Point 5: No effect sizes are reported for any statistical group comparison. Why not Especially since this is a pilot study, the effect sizes should be given.
Response 5:
Thank you for the comment. In a comparison of means, the effect size is given be the ratio of mean over standard deviation. As we reported these quantities for all comparison, we decided not to overload the presentation of results by additionally reporting their ratio. We agree that the information contained in the pilot study may be particularly relevant for the planning of further studies that aim to compare electrotherapy to other groups. We therefore added a sentence to the newly added Section 7 (Future Directions and Clinical Implications), saying: “The reported values for mean and standard deviations may be used in the sample size planning of future studies.”
Point 6: For example, I could not find degrees of freedom in group comparisons. Why were neither the test values nor the degrees of freedom given?
Response 6:
Thank you for the comment. We compared other papers published in Medicina and found that, in line with our manuscript, the presentation of descriptive statistics (e.g. mean and standard deviation) together with p-values is the standard way for reporting results of hypothesis tests. Indeed, the p-value directly reflects the test value and (in case of a t-test) the degrees of freedom. Regarding degrees of freedom for pairwise comparisons, these only apply to t-tests and F-tests and there essentially correspond to the sample size (e.g. the number of pairs minus 1 for paired t-tests). As the number of subjects is indicated in the text and in the Figures we feel that the relevant information is provided in concise form in the manuscript. We therefore decided not to separately report test values and degrees of freedom.
Point 7: Why did "only" five patients in the intervention group refuse to participate? This would have to be discussed intensively.
Response 7: As stated in the paper, patients had according to the ethics the right to quit without giving a reason for discontinuation; therefore, we can just assume the possible reasons. As now written in the paper, two patients quit the study after the first electrotherapy session, one patient after the 2nd, one patient after the 3rd, and one patient after the 4th session. Thereby, no adverse events were observed, respectively. In two of five patients, depression was diagnosed, and one patient had the diagnosis depression and anxiety disorder before the enrolment. NRS, QoL and ROM were investigated at the very beginning of the study before randomization was performed, and in the end of the study after 10 sessions of interventions; therefore, we can’t give any statement whether there were any changes in the pain level in between. A possible reason for discontinuation may have been a perceived inefficacy of therapy and mismatch with the patients’ therapeutic beliefs and expectations, which is now discussed in line 421-424 in the revised version.
Point 8: Since the groups seem to differ significantly on the pre-values, the pre-values should be considered as covariates in the context of an analysis of variance. In the pairwise comparisons presented in the manuscript, the pre-value is not taken into account in the analysis of the post-values.
Response 8:
Thank you for the comment. Indeed, including the pre-value in an ANCOVA type model can increase the statistical power and adjust for baseline imbalances; therefore, we have augmented the results section (please see line 253 – 291). However, the focus of the study is on descriptive statistics and on comparisons of pre versus post values, to assess changes within a group, since within subject comparisons are typically more powerful even at smaller sample size than between subject comparisons. For the within subject comparisons we reported, the pre values are naturally taken into account. Hypothesis tests for between group comparisons are, however, not in the focus of our analysis, as the small sample size would not allow for reasonable power and would also not be suitable to reliably fit more complex models that include covariates.

Reviewer 2 Report
The authors have developed a well-conducted and well-written study with the aim of analyzing the effect of regular suprathreshold electrotherapy in pain sensation and in cervical and lumbar range of motion, as well as in the subjective feeling of disability in daily life.
However, I suggest some clarifications or modifications that will in my opinion improve the quality of their manuscript:
1. I recommend ordering the Methodology section by sections. For example: 2.1. Study Design, 2.2. Sample, 2.3. Intervention, 2.4. Randomization and Blinding, 2.5. Outcomes Measurement, 2.6. Statistical Analysis
2. Please add the Ethics part also in the methodology section.
3. In the introduction / Discussion section, I recommend the authors to comment on prediction of patient satisfaction after treatment of chronic neck pain, mentioning the following reference: DOI: 10.3390/life13010048
4. To enrich the discussion, I recommend that the authors comment on the influence of expectations on the results of the treatment of patients with low back pain, as well as the beliefs of the health care providers who treat them: doi.org/10.47197/retos.v46.93950; DOI: 10.1186/s43161-022-00112-9
5. Could you remove the blue background from figure 2? It makes it difficult to understand the text it contains..
6. I find your comment about verbal suggestion in the Discussion section very interesting... I recommend commenting to the authors about the influence of verbal suggestion on the expectations of the study subjects supporting and discussing with reference to the following paper: DOI: 10.3390/ijerph18084206
7. In the Discussion section, could you add a section on "Future Directions and Clinical Implications"?
Author Response
Response to Reviewer 2 Comments
Point 1. I recommend ordering the Methodology section by sections. For example: 2.1. Study Design, 2.2. Sample, 2.3. Intervention, 2.4. Randomization and Blinding, 2.5. Outcomes Measurement, 2.6. Statistical Analysis
Response 1: Thank you for this suggestion. The methodology section is now restructured.
Point 2. Please add the Ethics part also in the methodology section.
Response 2: The ethics part is added as “2.2. Ethics and Clinical Registration” (line 76-80).
Point 3. In the introduction / Discussion section, I recommend the authors to comment on prediction of patient satisfaction after treatment of chronic neck pain, mentioning the following reference: DOI: 10.3390/life13010048.
Response 3: Thank you for this feedback. We decided to mention the paper in the discussion (line 413-415), but not in the introduction, because our study does not focus on anxiety and on the prediction of treatment satisfaction regarding a mobilization technique, but directly on NRS and on QoL after electrotherapy.
Point 4. To enrich the discussion, I recommend that the authors comment on the influence of expectations on the results of the treatment of patients with low back pain, as well as the beliefs of the health care providers who treat them: doi.org/10.47197/retos.v46.93950; DOI: 10.1186/s43161-022-00112-9
Response 4: We have added the expectation on the treatment results together with the potential effect of verbal suggestion in line 405-424.
Point 5. Could you remove the blue background from figure 2? It makes it difficult to understand the text it contains.
Response 5: The blue background is now removed.
Point 6. I find your comment about verbal suggestion in the Discussion section very interesting... I recommend commenting to the authors about the influence of verbal suggestion on the expectations of the study subjects supporting and discussing with reference to the following paper: DOI: 10.3390/ijerph18084206
Response 6: Thank you for the suggestion. Please see line 405-424 in the revised version of the manuscript.
Point 7. In the Discussion section, could you add a section on "Future Directions and Clinical Implications"?
Response 7: Thank you for the comment. It is now added as point 7 (line 493-502).

Round 2
Reviewer 1 Report
All my questions were answered, or the missing points were introduced. The manuscript can be published in this form.
Reviewer 2 Report
The authors have responded to all my observations.
I recommend the publication of their manuscript.